# Extracellular Matrix (ECM) Aging in the Retina: The Role of Matrix Metalloproteinases (MMPs) in Bruch’s Membrane Pathology and Age-Related Macular Degeneration (AMD)

**DOI:** 10.3390/biom15081059

**Published:** 2025-07-22

**Authors:** Ali A. Hussain, Yunhee Lee

**Affiliations:** 1Rescue, Repair and Regeneration Theme, UCL Institute of Ophthalmology, 11-43 Bath Street, London EC1V 9EL, UK; 2AltRegen Co., Ltd., 1201, Byoksan Digital Valley 5-cha, Beotkkot-ro, Seoul 08513, Republic of Korea

**Keywords:** matrix metalloproteinases (MMPs), extracellular matrix (ECM), Bruch’s membrane, aging, age-related macular degeneration (AMD)

## Abstract

The extracellular matrix (ECM) is a collagen-based scaffold that provides structural support and regulates nutrient transport and cell signaling. ECM homeostasis depends on a dynamic balance between synthesis and degradation, the latter being primarily mediated by matrix metalloproteinases (MMPs). These enzymes are secreted as pro-forms and require activation to degrade ECM components. Their activity is modulated by tissue inhibitors of metalloproteinases (TIMPs). Aging disrupts this balance, leading to the accumulation of oxidized, cross-linked, and denatured matrix proteins, thereby impairing ECM function. Bruch’s membrane, a penta-laminated ECM structure in the eye, plays a critical role in supporting photoreceptor and retinal pigment epithelium (RPE) health. Its age-related thickening and decreased permeability are associated with impaired nutrient delivery and waste removal, contributing to the pathogenesis of age-related macular degeneration (AMD). In AMD, MMP dysfunction is characterized by the reduced activation and sequestration of MMPs, which further limits matrix turnover. This narrative review explores the structural and functional changes in Bruch’s membrane with aging, the role of MMPs in ECM degradation, and the relevance of these processes to AMD pathophysiology, highlighting emerging regulatory mechanisms and potential therapeutic targets.

## 1. Introduction

This article presents a qualitative review focusing on extracellular matrix (ECM) aging and pathology in the retina, particularly Bruch’s membrane, and its role in age-related macular degeneration (AMD).

The ECM is a three-dimensional scaffold of macromolecular constituents that provides structural and functional support to the surrounding cells, allowing tissue organization, remodeling and serving as a communication platform between cells and their extracellular fluids [1,2,3]. Collagen species are the backbone of the matrix populated by other proteins, enzymes, proteoglycans, glycosaminoglycans, elastin, fibronectin and laminins in addition to the storage of growth factors [4,5,6]. Variations in the composition of these constituents dictate the type of ECM, reflecting its location and requirements for specific structural and functional demands [7,8,9].

The constituents of the ECM are prone to damage due to unfolding, denaturation, and oxidation, the degree of damage being dependent on the location of the matrix. Such alterations are expected to compromise the structural and functional aspects of the ECM [10,11,12]. The restoration of the ECM is fundamentally important and relies on the dynamic balance between matrix synthesis and its degradation, the latter being mediated by a family of proteolytic enzymes known as matrix metalloproteinases (MMPs) [13,14]. These metzincin-family proteases are initially secreted as inactive precursors (pro-enzymes) by the surrounding cells, and once activated, they possess the ability to break down most ECM constituents [15,16]. The maintenance of ECM functionality depends on precise coordination between its synthesis and degradation. This coordination is progressively lost with age, leading to the deposition of both normal and pathological ECM elements, reduced functional capacity, and eventual onset of degenerative alterations. To elucidate the mechanisms that drive aging and its progression to pathological states, it is essential to study an appropriate ECM model. Bruch’s membrane serves as a valuable model for studying aging, as it can be obtained from human donor eyes in the form of intact sheets, enabling comprehensive structural, biochemical, and functional analyses. These investigations help elucidate the mechanisms underlying the shift toward pathology, particularly those implicated in AMD.

### Bruch’s Membrane as an Ideal Model of an ECM

Key components of the visual unit include photoreceptors, the retinal pigment epithelium (RPE), which functions as the blood–retinal barrier, Bruch’s membrane, made up of ECM elements, and the fenestrated vasculature supplying the retina (Figure 1a). Photoreceptors are the light detectors with rhodopsin molecules embedded in discs that are stacked in the outer segment. Light detection and subsequent phototransduction result in neurotransmitter modulation at the synaptic pedicle.

Bruch’s membrane is a penta-laminated ECM structure (Figure 1b) comprising a central elastic layer [C] composed of elastin, flanked by outer [B] and inner [D] collagenous layers consisting of collagen types I, II and V. These layers are sandwiched between the two basement membranes of the RPE [A] and the choriocapillaris [E], both containing collage type IV, fibronectin, and laminin [17,18]. Lying between the RPE and the fenestrated capillaries of the choriocapillaris, Bruch’s membrane mediates the inward delivery of nutrients and antioxidants to support the intense metabolic activity of photoreceptors and outward removal of fluids and toxic waste products.

The aging of Bruch’s membrane is characterized by a range of structural and compositional changes, including increased membrane thickness [19], accumulation of both physiological and abnormal ECM components [10], and the enhanced formation of oxidative and enzymatic cross-links that contribute to the generation of advanced glycation end-products [20]. In addition, lipid-rich deposits gradually accumulate within the membrane [21]. Figure 2 provides a schematic summary of these age-related and pathological changes in Bruch’s membrane. During normal aging, expansion of the ECM suggests impaired degradation processes, ultimately leading to diminished transport capacity needed to sustain the RPE and photoreceptors.

The varying thickness of the double-headed arrows in Figure 2 illustrates the transport efficiency of Bruch’s membrane. In youth, the membrane remains highly permeable, enabling effective exchange of nutrients and the clearance of metabolic waste and toxic byproducts, thereby supporting photoreceptor health. With aging, there is progressive accumulation of both intact and damaged ECM components, along with toxic substances originating from photoreceptors and the RPE. In normally aging individuals, such alterations impair the transport function of Bruch’s membrane, which is often reflected in delayed dark adaptation. In more advanced aging, particularly in the context of AMD, transport becomes severely restricted. This leads to a metabolic deficit that, through inflammatory mechanisms, contributes to the degeneration and eventual loss of both RPE cells and photoreceptors, ultimately resulting in vision loss.

In advanced stages of aging associated with AMD, a significant metabolic disturbance occurs, which, through the activation of inflammatory pathways, ultimately results in the loss of RPE and photoreceptors, leading to irreversible blindness. AMD continues to be the leading cause of untreatable vision loss among the elderly [22]. The prevalence of this condition increases with age, ranging from approximately 1.3% to 3% in individuals younger than 55 years to as high as 28% to 30% in those aged 75 and above [23]. Clinically, AMD is classified into early and late stages. Early AMD is marked by the presence of drusen, lipid-rich deposits overlying Bruch’s membrane, as well as pigmentary alterations in the fundus, such as areas of hypo- or hyperpigmentation [24]. As the disease progresses to its late stage, it may result in geographic atrophy of the RPE followed by degeneration of the photoreceptors, a condition referred to as ‘dry’ AMD. The advanced phase may also involve complications such as pigment epithelial detachments (PEDs, affecting 12–20% of patients) and neovascularization (occurring in 10–20% of cases), the latter constituting what is clinically termed ‘wet’ AMD. Although neovascularization can be managed to an extent with anti-VEGF intervention, the underlying degenerative phase remains unabated. Thus, currently, there is no treatment option for the majority of AMD patients.

The remainder of the review will address the mechanisms underlying these aging changes in the ECM of Bruch’s membrane regarding the structural and functional alterations, the inability of the MMP rejuvenation system to combat these changes and extend these results to formulate potential therapeutic interventions to address AMD.

## 2. Compositional Alterations in the ECM of Aging Bruch’s Membrane

In the photoreceptor, the high level of mitochondrial oxidative phosphorylation, the presence of high oxygen tension and polyunsaturated fatty acids (PUFAs), high level of toxic retinoids, and light, result in an explosive environment generating extensive oxidative and free-radical damage [25,26,27].

There are three major processes within the photoreceptor cell that lead to the production of toxic metabolites, that after limited processing in the RPE, are extruded onto Bruch’s membrane (Figure 3) [28,29]. The first is mediated by the released chromophore following rhodopsin activation, namely, all-trans retinal (AT-RL) [28,29]. Second, and perhaps the most important, is the photo- and free-radical oxidation of polyunsaturated fatty acids (PUFAs) leading to degradation and fragmentation producing a variety of highly reactive carbonyl compounds [30,31]. Third, is the production of lipid and lipo-protein aggregates and carbonyl adducts on proteins leading to the formation of highly cytotoxic compounds [30,32].

Photoreceptors are a key origin of damage, producing reactive substances that chemically alter lipid and protein structures, leading to the formation of lipid–lipoprotein aggregates and cytotoxic proteins [57]. These damaged discs are transferred to the RPE and undergo further damage that impairs lysosomal-mediated degradation. Damaged material that cannot be broken down is either stored as lipofuscin granules or extruded onto Bruch’s membrane. Damage to Bruch’s membrane causes the accumulation of normal and abnormal ECM products to compromise its transport functions to support the RPE and photoreceptors [57,58].

Following rhodopsin activation, released AT-RL can react with phosphatidyl ethanolamine to form the adduct retinylidene-phosphatidylethanolamine (NRPE) and this can react with a second AT-RL molecule leading to the irreversible formation of a bis-retinoid [33]. Photo-oxidation of these bis-retinoids leads to the formation of epoxides and endoperoxides and their degradation leads to the production of methylglyoxal and glyoxal and these oxo-aldehydes complex with and damage nearby proteins [41,42].

Polyunsaturated fatty acids (PUFAs), particularly docosahexaenoic acid (DHA), which constitutes approximately 80% of the fatty acids in photoreceptor outer segments, are highly vulnerable to free-radical-mediated oxidation [43]. This lipid peroxidation often results in molecular fragmentation, producing a diverse array of secondary products such as malondialdehyde (MDA), acrolein, 4-hydroxy-2-nonal (HNE), etc. These reactive byproducts readily form covalent bonds with proteins, leading to alkyl- and carboxyalkyl-pyrrole modifications [37,38]. Consequently, the oxidative degradation of disc PUFAs promotes the accumulation of lipid aggregates, lipid-protein adducts, protein cross-links, and carboxy-ethyl-pyrrole-(CEP)-protein adducts.

As part of the physiological disc shedding mechanism, oxidative damage originating in the photoreceptor is passed on to the RPE. Given that the RPE exists in a similarly oxidative environment, the harmful reactions initiated in the photoreceptor persist within the RPE phagosome.

Fusion of the phagosome with RPE lysosomes forms the phago-lysosomal particle initiating the attempted breakdown of engulfed disc membranes. Lysosomal enzymes are responsible for breaking down normal, unmodified proteins and lipids within the phago-lysosome, allowing their basic metabolic components to be recycled back to photoreceptor cells. Yet, structurally altered proteins, lipid-derived conjugates, cross-linked proteins generated through lipid–carbonyl interactions, and aggregated lipid forms that resist enzymatic degradation tend to accumulate within the phago-lysosomal compartment [39]. The degradation of bis-retinoids within lysosomes leads to the formation of A2E, a key fluorophore associated with aging [45]. A2E and related bis-retinoid compounds are prone to further oxidative reactions, giving rise to multiple reactive products, including epoxides, cyclic peroxides, furanoid oxides and carbonyls [34,35]. These bis-retinoid derivatives not only compromise lysosomal enzyme function but also impair lysosomal membrane integrity, resulting in proton pump inhibition. This in turn elevates the pH, further reducing enzymatic activity [36]. Many of the damaged lipoproteins and protein aggregates become entrapped with bis-retinoids and are stored as lipofuscin, an autofluorescent pigment enclosed in membrane-bound vesicles [46,47]. The volume of lipofuscin increases with age and may occupy up to 20% of the cytoplasm in aged RPE cells [48].

The extruded toxic and undigested material from the RPE undergoes further oxidative modifications leading to cross-linking and oxidative reactions that damage the ultrastructural architecture of Bruch’s membrane [11,40]. Lipid constituents entering Bruch’s membrane will undergo free-energy-driven association and aggregation leading to the accumulation of 100 nm diameter lipid-rich particles observed in the inner collagenase layer of Bruch’s membrane [44]. Thus, in addition to the toxic metabolites mentioned, deposits within Bruch’s membrane also contain phospholipids, triglycerides, cholesterol, cholesterol esters, unsaturated fatty acids, peroxidized lipids and apolipoproteins, vitronectin, immunoglobulins, amyloid, and complement [21,49,50,51]. Damaged collagen constitutes about 50% of the total collagen content in Bruch’s membrane from aged donor eyes [10]. Also present are deposits of iron and zinc that not only promote oxidative damage but may also stabilize the debris within Bruch’s membrane [52].

Particulate debris discharged onto Bruch’s membrane may become trapped or transported out into the choroidal circulation. Originally, Bruch’s membrane was thought of as a simple rigid membrane filter and in vitro experiments showed that it could transport proteins in the molecular weight range 40–200 kDa and dextrans up to 500 kDa [12,53]. Subsequent work in vivo showed that much larger complexes (such as lipoproteins and microspheres) could also be transported [54,55]. This suggests that far from being rigid, Bruch’s membrane can undergo compression and decompression cycles in response to blood pulsations in the large choroidal vessels underlying the membrane. Such cyclic perturbations allow the entry of larger molecules and removal of membranous material deposited by the RPE. Compression–decompression cycles can therefore be regarded as a self-cleansing mechanism that is dependent on the elastic nature of the membrane. As aging progresses, Bruch’s membrane loses elasticity, becomes more rigid, and its self-cleansing capacity diminishes, resulting in increased debris deposition [56].

It should be noted that the oxidative reactions and damage occur despite the presence of an impressive antioxidant machinery in both the photoreceptor cell and the RPE (vitamins C and E, macular pigments, and an array of enzymes such as catalase, peroxidase, and superoxide dismutase) [59,60,61].

The extensive structural alterations that occur in Bruch’s membrane during normal aging are likely to impair its functional properties.

## 3. Functional Alterations in the ECM of Aging Bruch’s Membrane

Photoreceptors exhibit exceptionally high levels of oxidative metabolism, among the most intense of any cell type in the body, and thus depend on a continuous supply of glycolytic substrates and antioxidant components to be transported across Bruch’s membrane and the RPE [62,63]. In parallel, fluids and metabolic waste are cleared in the reverse direction toward the choroidal vasculature for elimination.

### 3.1. Fluid Transport Across Bruch’s Membrane: Concept of Failure Thresholds

Fluid originating within the retina arising from both the inner retinal capillary network and retinal metabolic processes is cleared primarily by the RPE [64]. Approximately 70% of this fluid efflux is facilitated through active transport mechanisms, which are driven by the movement of ions from the apical to the basal surface of the RPE [65,66]. Although the exact rate of fluid clearance by the RPE in the human eye remains undetermined, one clinical study estimated this value by monitoring the resolution of a retinal bleb in a case of non-rhegmatous detachment via B-scan ultrasonography. Assuming RPE-mediated removal, the rate was calculated to be approximately 0.11 μL/h/mm^2^ [67]. Consistent with these findings, various animal studies utilizing both experimentally induced retinal blebs and in vitro Ussing chamber systems have reported similar fluid transport rates, averaging around 0.13 ± 0.11 μL/h/mm^2^ [64,68,69,70].

After being actively transported by the RPE, fluid must pass through Bruch’s membrane to reach the choroidal circulation for elimination. The minimum level of hydraulic conductivity required of Bruch’s membrane to accommodate the daily volume of fluid cleared by the RPE has been estimated at 0.65 × 10^−10^ m/sec/Pa. This value represents the threshold below which fluid transport becomes insufficient, commonly referred to as the failure point [18,71]. A fall in hydraulic conductivity of Bruch’s membrane below this failure threshold will result in fluid pooling on top of the membrane leading to an RPE detachment, an occurrence observed in about 12–20% of AMD patients [72]. In this situation, the increased diffusional distance between Bruch’s membrane and the RPE will diminish the transport rate of essential nutrients posing a threat to photoreceptor survival. To evaluate the potential risk of this occurrence, measurements of hydraulic conductivity in Bruch’s membrane have been conducted using macular tissue samples from normal human donors across a wide age spectrum, ranging from 9 to 91 years (Figure 4) [71,73,74].

Hydraulic conductivity of Bruch’s membrane was assessed using tissue from 56 human donors ranging in age from 9 to 91 years. Semi-logarithmic analysis of the data demonstrated an age-dependent exponential decline in conductivity, with a calculated half-life of approximately 16 years. The dashed line in Figure 4 represents the failure threshold, defined as the minimum conductivity required to facilitate the clearance of fluid exported by the RPE. In younger and middle-aged donors, Bruch’s membrane exhibits a notable excess capacity for fluid transport. In contrast, with increasing age, this capacity steadily declines, gradually approaching the threshold below which fluid clearance becomes insufficient (modified from Refs. [71,73,74]).

The plot shows that for the major portion of the human life-span, the hydraulic conductivity of Bruch’s membrane is much elevated compared to the failure threshold. However, hydraulic conductivity shows an age-related exponential decrease, with a half-life of approximately 16 years. In other words, the conductivity is reduced by half every 16 years of life. Therefore, in the elderly, there is a heightened risk of failure, and as shown in the plot, some elderly control subjects have already drifted towards the failure threshold. AMD donors showed reduced hydraulic conductivity compared to age-matched controls indicative of faster deterioration in fluid transport across Bruch’s membrane in this disease [12].

### 3.2. Metabolite Transport Across Aging Bruch’s Membrane

Metabolites span a broad spectrum, from low molecular weight compounds such as sugars and amino acids to high molecular weight protein-bound carrier complexes (hydrodynamic radii of 3–12 nm), and are transported passively down concentration gradients across Bruch’s membrane governed by the following simple diffusion equation:J = D × ΔC/L
where the diffusional flux per unit area (J) is proportional to the diffusion coefficient (D) of the molecule and the concentration difference across the membrane (ΔC) is inversely proportional to the diffusion path length (L).

The rough doubling in the thickness of Bruch’s membrane over the human life-span would dictate that the diffusional fluxes should be halved over the same period. If the size of the traversing molecule is much smaller than the spaces in the fibrous meshwork of Bruch’s membrane, then its diffusional coefficient within the membrane will be similar to its value in bulk solution. Amino acids are classified among the small molecular weight solutes, and studies have shown that the transport of hydrophilic and aromatic amino acids, such as glycine and phenylalanine, declines significantly with age. Between the ages of 0 and 90 years, flux levels for glycine and phenylalanine were reduced by approximately 52% and 70%, respectively [75]. This age-related reduction in flux is compatible with the 2–3-fold increase in the thickness of Bruch’s membrane.

Albumin, along with protein carriers for retinol, trace metals, vitamins, and lipoproteins, exhibits hydrodynamic radii in the range of approximately 3 to 12 nm. These larger molecular species are more likely to interact with the structurally altered and denatured components of Bruch’s membrane during aging, resulting in a significant reduction in their diffusional transport rates [12].

To allow comparison across studies conducted at different solute concentrations, researchers have used a concentration-independent parameter known as the permeability coefficient (P), which is defined as the flux per unit area (J) divided by the difference in concentration (ΔC).

Serum proteins typically range in molecular weight from 40 to 200 kDa, with albumin, at around 65 kDa, being the most prevalent. Measurements of protein permeability across the macular region of human Bruch’s membrane indicate a marked decline with age. The average permeability value decreases from 3.5 × 10^−6^ cm/s in the first decade of life to 0.2 × 10^−6^ cm/s by the ninth decade, representing a reduction of more than tenfold [53].

In a similar manner, the diffusion of albumin, which has a molecular weight of 65 kDa and a hydrodynamic radius of 3.5 nm, across the macular region of Bruch’s membrane was found to decrease exponentially with age. This decline follows a half-life of approximately 18 years and results in an almost tenfold reduction in albumin diffusion in elderly individuals (Figure 5) [76]. The precise threshold at which the diffusion of carrier-sized proteins becomes insufficient is not yet established. Despite this, elderly individuals without overt disease often exhibit impaired dark adaptation, which has been attributed to reduced retinol transport across Bruch’s membrane [77]. Thus, the diffusional status of these subjects has been taken as the failure threshold.

Albumin with MW 65 kDa and hydrodynamic radius of 3.5 nm was chosen as a test probe representing the size of most carrier proteins that cross Bruch’s membrane delivering essential metabolites to the photoreceptors. The impact of aging on diffusion across Bruch’s membrane was evaluated in tissue obtained from 33 control donors between the ages of 12 and 92 years. The ability of albumin to diffuse through the membrane declined in an age-dependent exponential manner, with a calculated half-life of approximately 18 years. Since elderly control subjects show problems with dark adaptation due to inefficient delivery of carrier-mediated retinol, the diffusional capacity in elderly donors was designated as the failure threshold (modified from Ref. [76]).

As a result, elderly individuals tend to approach the failure threshold for diffusion. In donors with AMD, the rate of diffusional transport across Bruch’s membrane was significantly lower than that observed in age-matched controls [12]. This decline in transport efficiency is believed to impose metabolic stress on the RPE and photoreceptors, potentially triggering the onset of pathological changes.

The age-related reduction in diffusional potential across Bruch’s membrane has ramifications also for the clearance of debris leading to an exponential increase in lipid deposition and the age-pigment A2E [21,78].

## 4. Abnormalities of the MMP System of Bruch’s Membrane in Aging and Disease

Aging of Bruch’s membrane is associated with gross accumulation of damaged ECM material that should be amenable to degradation by the MMP machinery present. In the following sections, the mechanisms for diminished MMP activity in aging Bruch’s membrane are examined. This entails an assessment of the level of MMP species present, their covalent and non-covalent modification and sequestration, and problems with the MMP activation mechanism, all in relation to aging.

ECM degradation is regulated by a group of metalloproteinases belonging to the metzincin superfamily of endopeptidases. This superfamily includes matrix metalloproteinases (MMPs), a disintegrin and metalloproteinase group (ADAMs), and ADAMs with thrombospondin motifs (ADAMTSs) [79,80]. These enzymes not only act on ECM components but also target a variety of non-matrix substrates, such as signaling molecules, inflammatory mediators including cytokines and chemokines, as well as receptors and cell-adhesion molecules [81,82,83,84].

In humans, the MMP family consists of 24 members, which are categorized based on their substrate specificity. These include gelatinases (MMPs 2 and 9), stromelysins (MMP 3 and 10), collagenases (MMP 1, 8 and 15), and membrane-type MMPs (MT-MMPs 1–6). The membrane-associated MT-MMPs possess broad substrate specificity and are known to degrade fibrillar collagens (type I, II, and III), laminins 1 and 5, fibronectin, and vitronectin [85,86,87,88].

MMPs and ADAMTS are typically secreted into the extracellular environment as inactive or latent pro-enzymes, requiring the proteolytic cleavage of a short pro-peptide segment for activation. Once activated, MMPs exhibit the capacity to degrade a wide range of ECM components [15,89,90,91].

### 4.1. Activation of Pro-MMPs

A conserved cysteine in the N-terminal pro-domain of pro-MMPs binds to the catalytic zinc ion, forming an S-Zn^2+^ interaction that blocks substrate binding and maintains enzyme latency. Activation is initiated by a protease that cleaves a small peptide from the N-terminal of the pro-domain breaking the S-Zn^2+^ link, also known as the cysteine-switch. An autocatalytic step then follows to remove the entire pro-domain leading to a fully activated MMP enzyme, with a loss in molecular weight of about 8–12 kDa [92]. Chemical agents such as sodium dodecyl sulphate (SDS) can induce conformational shifts in pro-MMPs that displace the pro-domain. Subsequent treatment with Triton-X100 yields a partially active enzyme, despite no change in molecular weight, enabling the identification of pro-MMPs by zymography. In addition to chemical induction, pro-MMPs can also undergo allosteric activation through interactions with ECM components and cell surface receptors [93,94,95,96].

### 4.2. MMP Species in Bruch’s Membrane

The secretion of pro-MMPs 1, 2, 3, and 9 by RPE and choroidal cells has been confirmed, along with their localization within Bruch’s membrane [97,98,99,100,101]. Additionally, tissue inhibitors of metalloproteinases (TIMPs) have also been identified in this region. TIMPs 1 and 2 exhibit free mobility within Bruch’s membrane, whereas TIMP3 is more firmly associated with the ECM and remains tightly bound to its structural components [98,102]. In Bruch’s membrane, the homeostatic enzyme for degradation of the matrix is MMP2 with MMP9 being the inducible form. Gelatinase species (such as pro-MMPs 2 and 9) can also undergo post-translational changes including methylation, phosphorylation, acetylation, oxidation and glycosylation [103,104,105,106,107]. These changes can modify protein function due to alterations in structure that affect protein–protein interactions and degradation activity [108,109,110,111]. Several charge variants are known for pro-MMPs 2 and 9 but none have yet been investigated in Bruch’s membrane [112,113,114,115]. Theoretically, these charge isoforms can exhibit variable binding affinities toward substrates, receptors, and inhibitory molecules but the effect of these changes on proteolytic activity has not yet been determined [115,116]. While RPE cells are a known source of MMPs, particularly MMP2, the possibility that circulating MMPs from plasma may cross the Bruch’s membrane via fenestrated capillaries should not be excluded. Age-related increases in plasma MMP2 have been reported, and their translocation into the subretinal space remains a plausible mechanism requiring further investigation [117].

### 4.3. Modification of Gelatinase Species in Bruch’s Membrane

Gelatinase enzymes present in Bruch’s membrane are subject to both covalent and non-covalent modifications. These alterations contribute to their aggregation and influence their distribution between free and matrix-associated compartments. On extraction of the membrane with SDS-containing buffers for zymography, the non-covalently attached species will be released and aggregated species will be disintegrated resulting in a non-physiological profile of the MMP species present. In the zymogram of Figure 6, the major gelatinase species of Bruch’s membrane present were pro-MMPs 2 and 9 (MW 65 kDa and 92 kDa, respectively), together with their activated forms. In addition, several gelatinase species of MW greater than 140 kDa were also documented [97,101,118,119,120,121]. Two of these high molecular weight (HMW) species have been identified as HMW1 (MW 164 kDa) and HMW2 (MW 344 kDa) and are mono or heteropolymers of pro-MMPs 2 and 9 [122,123]. Both HMW1 and HMW2 were susceptible to degradation by amino phenyl mercuric acetate (APMA), suggesting that the individual subunits were linked through disulfide bonds. Degradation of HMW1 showed it to contain pro-MMP9 whereas HMW2 was a polymer of pro-MMPs 2 and 9 [122].

Zymograms are standard, 10% SDS-PAGE gels that have incorporated into them an MMP substrate, gelatine (1%). Electrophoresis separates MMPs according to molecular weight, with the smallest running towards the bottom of the gel. Coomassie blue stains the whole gel except the MMP bands because in these regions, the gelatine has been hydrolyzed. The major MMP species in Bruch’s membrane were the pro-MMPs 2 and 9 together with their activated forms. Higher molecular weight species were routinely present and two of these have been termed HMW1 and HMW2.

The level of free and bound MMP species together with their native configuration can be determined if Bruch’s membrane is fractionated. Homogenization of Bruch’s membrane in physiological buffers followed by centrifugation provides a pellet containing the structural elements of the matrix (with bound MMP species) and a supernatant housing the free components. Zymographic profiling of the pellet fraction enables the assessment of matrix-bound MMP species, while gel-filtration chromatography of the supernatant permits the resolution of free MMP complexes under native conditions.

Elution of the supernatant fraction was marked by a large peak at the leading edge of the elution profile representing a complex of molecular weight exceeding 4000 kDa. Zymography of this peak showed it to contain predominantly HMW2, pro-MMP9, and HMW1 with a trace amount of pro-MMP2, this complex being designated as the large macromolecular weight MMP complex (LMMC) [122,123]. This large complex therefore sequesters MMP species (within the free compartment) and is expected to reduce the porosity of the matrix affecting the diffusional status of traversing molecules. Later, elution of free HMW1, HMW2, pro-MMPs 2 and 9 were subjected to an analysis of MW and showed that HMW2 and pro-MMP2 existed in the soluble fraction as monomers whilst HMW1 and pro-MMP9 were present as dimers.

### 4.4. Aging Changes in the Content of MMP Species of Bruch’s Membrane

If samples of Bruch’s membrane are perfused or vortexed, the free MMP species are released whereas the bound and LMMC particle (which is too large to exit the membrane) remain in the tissue sample. The age-related distribution of MMPs between bound and free forms was assessed. This analysis was performed using tissue obtained from ten human donors ranging in age from 21 to 84 years [124]. The percentage of total MMPs 2 and 9 in the bound state increased exponentially with age, *p* < 0.01. Thus, between the ages of 21 and 84 years, bound pro-MMP2 increased from 1.3% to 15.6% and pro-MMP9 from 9.4% to 62.1%. Active MMP2 was present in 6 of the 10 eyes examined and the amount bound was 21%. The proportion of HMW1 associated with the bound fraction increased with age, rising from 17 percent at 21 years to 53 percent by 84 years. In contrast, the majority of HMW2 remained in the bound state throughout the entire age range studied.

The various transformations of pro-MMPs 2 and 9 and the binding and sequestration of the species (as described above) can be configured as the MMP pathway of aging (Figure 7). In this pathway, age-related changes of Bruch’s membrane is associated with increased covalent modification of pro-MMPs 2 and 9 to form HMW1 and HMW2 and their binding to the matrix and incorporation into the LMMC particle. Matrix degradation, and hence its rejuvenation, requires the pathway to move in the opposite direction.

Pro-MMPs 2 and 9 are the major free MMP species and their activation results in the rejuvenation of Bruch’s membrane. Nevertheless, with advancing age, free pro-MMPs 2 and 9 become covalently modified, generating HMW1 and HMW2, which subsequently contribute to the formation of the LMMC particle (MW > 4000 kDa). This sequestration of pro-MMPs reduces the free amount for the activation process contributing to the aging of the matrix of Bruch’s membrane.

### 4.5. Aging and the Matrix Degradation Potential in Bruch’s Membrane

Despite the presence of the MMP system, the ECM of aging Bruch’s membrane shows impaired degradation with the accumulation of oxidized, cross-linked and denatured collagen. MMPs, including MMP2, play a dual role: contributing to the maintenance of the ECM and also potentially facilitating the progression of AMD [125]. Traditionally, the general consensus has been that it is the ratio of activated MMPs to TIMPs that dictates the degradative potential in the matrix. The finding that the inhibitor TIMP3 was elevated in aging Bruch’s membrane would suggest that the diminished ratio of active-MMP/TIMP3 may contribute to reduced enzyme activity [126,127]. However, zymographic analysis (which separates active MMPs from their inhibitors) showed that the actual levels of active MMP2 present were reduced on aging [97]. Although pro-MMP levels increase with age, the concentration of active-MMP species has been found to decline. In individuals with advanced AMD, the activity of MMP2 and MMP9 was reduced to approximately 50 percent of that observed in age-matched controls [123]. Additional factors contributing to this decline in enzymatic activity include an age-associated rise in intermolecular collagen cross-linking and the accumulation of AGEs and ALEs. These modifications reduce the susceptibility of collagen to proteolytic degradation and strongly suppress MMP function [128,129,130,131,132]. Moreover, trace metals such as zinc and iron have been shown to modulate the activity of MMP2 and MMP9, and changes in their local concentrations with aging may further influence the proteolytic balance in Bruch’s membrane [133].

In addition to enzymatic regulation, microRNAs (miRNAs) act as post-transcriptional modulators of MMPs, influencing ECM remodeling. For example, miR-21 has a protective effect on angiogenesis by reducing cell death and promoting cell survival and migration, in part through targeting TIMP3 and regulating MMP2 and MMP9 [134]. miR-29b has been identified as a potential regulator of MMP2 [135]. Other miRNAs, such as miR-204, are also involved in RPE homeostasis [136]. Moreover, modulation of specific miRNAs, such as overexpression of miR-21, miR-31, or miR-150, or silencing of miR-23 and miR-27, has been shown to attenuate choroidal neovascularization (CNV) in laser-induced models [137]. These findings suggest that miRNAs function as endogenous regulators of MMP-dependent ECM remodeling in AMD pathogenesis.

### 4.6. Problems with the Activation of Pro-MMP2 in Aging and Disease

The most important parameter for degradation of the ECM of Bruch’s membrane is the level of activated-MMP2. Activation of pro-MMP2 occurs on the basolateral surface of the RPE and is shown in schematic form in Figure 8. MMP14, a transmembrane matrix metalloproteinase located on the basal surface of the RPE, interacts with TIMP2 to form a binary complex. This complex serves as a docking site for free pro-MMP2, resulting in the formation of a ternary complex. Activation occurs when a second MMP14 molecule cleaves the pro-domain of pro-MMP2, thereby generating the active form of MMP2 [138,139,140]. Thus, the activation process requires sufficient amounts of free TIMP2 and pro-MMP2 and good mobility within the matrix to interact with MMP14 on the basal surface of the RPE.

Activation of pro-MMP2 occurs at the basolateral surface of the RPE, where the transmembrane enzyme MMP14 first binds to TIMP2, forming a binary complex. This complex facilitates the recruitment of pro-MMP2, resulting in the formation of a ternary complex. A second MMP14 molecule then cleaves the pro-domain of pro-MMP2, producing the active enzyme. The efficiency of this activation process depends on adequate levels of free pro-MMP2 and TIMP2, as well as their mobility within the matrix to ensure effective interaction with MMP14.

Although the total level of pro-MMP2 is elevated in Bruch’s membrane with aging, the free level is compromised due to competing reactions for this species as outlined in Figure 9. Competitive reactions removing free pro-MMP2 include covalent modification with pro-MMP9 to produce HMW2, complexation with pro-MMP9, HMW1 and HMW2 to produce the LMMC particle, and sequestration due to binding to the matrix architecture. Lack of activated MMPs is not the only reason for reduced degradation capacity in the membrane. These activated species are also trapped in the overcrowded aging matrix and are not mobile enough in the grossly altered aged matrix to interact with their substrates. When Bruch’s membrane from aged donors is placed in Ussing chambers and exposed to physiological buffer solutions, there is very little (if any) release of activated forms. However, when the debris in Bruch’s membrane is dispersed by amphipathic steroidal glycoside molecules (saponins), the trapped activated MMPs are released allowing degradation [76].

The reduction in free pro-MMP2 occurs due to binding to the matrix, covalent polymerization with pro-MMP9 to form HMW2, and incorporation into the LMMC particle. The activated MMP can also be trapped or bound in the matrix.

In AMD, the covalent interaction between pro-MMPs 2 and 9 to form HMW2 is accelerated due to elevated levels of pro-MMP9 in both plasma and Bruch’s membrane, the net result being a reduction in the level of pro-MMP2 available for the activation process [123,141]. Thus, in Bruch’s membrane of AMD donors, the free level of pro-MMP9 was increased nearly three-fold (*p* < 0.005) and a reduction in the level of free pro-MMP2 to 24% of control [123]. This is most likely to be responsible for the 50% reduction in activated MMP2 levels in AMD.

## 5. Therapeutic Potential to Modulate the Age-Related Deterioration of the ECM

As outlined earlier, the primary source of damage in the visual unit is the oxidative stress in the photoreceptor and RPE with the toxic metabolites generated damaging the structural components of Bruch’s membrane. Antioxidant measures provide one possibility for intervention. Alternatively, the removal of debris from Bruch’s membrane may allow greater mobility of TIMP2 and pro-MMP2 for the activation process. Another possibility is to increase the secretion of activated MMP2 from the RPE itself. These considerations for therapeutic intervention are briefly described below.

### 5.1. Antioxidant Supplements for AMD

Given that oxidative stress originating in photoreceptors is a key driver of subsequent damage to the RPE and Bruch’s membrane, antioxidant supplementation has been proposed as a preventive strategy for AMD [142,143,144]. Although the AREDS formulation has been widely used for more than two decades, its efficacy remains a subject of debate, as it has not been conclusively shown to prevent legal blindness in advanced stages of the disease [145,146,147]. With hindsight, the greater than 10-fold reduction in diffusion capacity in elderly Bruch’s membrane and more so in AMD is not expected to increase the transport of supplements to any extent to justify their use. In addition, current supplementation strategies fail to address the decline in reverse transport across Bruch’s membrane, particularly the clearance of toxic metabolites that are believed to initiate neovascular or inflammatory responses.

### 5.2. Improving the MMP Machinery in Bruch’s Membrane

Improving the transportation pathways across Bruch’s membrane would improve the bi-directional traffic increasing the inward delivery of nutrients and antioxidants and outward removal of toxic metabolites. This requires addressing the MMP machinery of aging Bruch’s membrane.

Given that TIMP3 levels are raised in aging Bruch’s membrane and the presence of potent inhibitors of MMPs such as cross-linked collagen and AGEs and ALEs, would the enhanced presence of activated-MMP2 serve to improve the degradative potential of the membrane? When purified and activated MMP2 was incubated with donor human Bruch’s membrane, the transport characteristics of the preparation were much improved implying improved degradation of the ECM [148]. Thus, the presence of endogenous MMP inhibitors was without effect on the proteolytic activity of activated-MMP2. A limitation is the absence of in vivo studies evaluating the effects of recombinant active MMP2 administration on Bruch’s membrane integrity and visual function. This represents a key area for future investigation.

#### 5.2.1. Improving the Mobility of Pro-MMP2 and TIMP2 for Activation

One of the barriers to the natural activation of pro-MMP2 is the low mobility within the aging matrix for interaction with MMP14. Improving mobility within the matrix would require the removal of lipid-rich proteinaceous debris to allow the greater activation of pro-MMP2 and release of trapped activated-MMP2.

Saponins are amphipathic tri-terpenoid glycosides possessing both hydrophilic and hydrophobic regions, which allow them to readily interact with and integrate into lipid-based structures such as liposomes, micelles, biological membranes, and lipid deposits, thereby facilitating their dispersal [149,150,151,152]. When saponins were perfused through donor human Bruch’s membrane, they were found to remove multiple lipid classes, release the high molecular weight species HMW1 and HMW2, and liberate sequestered active MMPs. These effects collectively contributed to reinitiating matrix remodeling processes [76]. A single perfusion with saponins led to substantial enhancement in both the hydraulic conductivity and diffusional capacity of Bruch’s membrane.

As mentioned earlier, the key functional biomarker in AMD is the delayed recovery of rod photoreceptor sensitivity after exposure to a bright flash of light [153,154,155]. Briefly, the flash bleaches rhodopsin, generating opsin and all-trans retinal, which leads to a loss of sensitivity. Recovery requires the delivery of 11-cis-retinal from retinoid stores in the RPE back to the photoreceptors to regenerate rhodopsin. In AMD, this process is slowed due to reduced retinoid levels in the RPE, which result from impaired transport of retinol. Retinol is carried as a complex with retinol binding protein and transthyretin, with a combined molecular weight of approximately 75 kDa, and this complex cross Bruch’s membrane inefficiently in AMD. If saponins could improve transport across Bruch’s membrane, then retinoid stores in the RPE would be replenished resulting in improved dark adaptation. A double-blind placebo controlled clinical trial was undertaken and showed that a 200 mg oral daily dose of saponins could reverse the recovery rate of dark adaptation in AMD patients to control levels within 4 months of treatment (Figure 10) [156]. After two months of treatment, subjects in the placebo group (open circles) exhibited negligible or no changes in S2 gradients. In contrast, all individuals with AMD receiving saponin supplementation (filled circles) demonstrated significant improvements in S2 gradients (*p* < 0.005). By the four-month mark, the S2 gradients in the placebo group remained stable, showing no meaningful change. Notably, four AMD patients who continued saponin administration for the full duration experienced further improvement relative to their two-month values (*p* < 0.01). This improvement in the transport of retinol means that the transport of all other carrier-mediated nutrients, antioxidants, vitamins and essential metals might also be improved, constituting a possible treatment option for AMD. Similarly, improved transport across Bruch’s membrane would facilitate the removal of toxic metabolites.

#### 5.2.2. Enhancing the Levels of Activated MMPs 2 and 9 Using Lasers

When RPE cells are cultured, their proliferation is associated with the release of copious amounts of activated MMPs [148]. Similarly, if RPE cells are plated onto donor human Bruch’s membrane and allowed to proliferate to confluence, the result is improved transport properties of Bruch’s membrane, assumed to be mediated by the release of activated MMPs (Figure 11) [91]. Recent laser advancements allow precise RPE ablation without harming nearby photoreceptors, triggering regenerative RPE migration and the release of activated MMPs.

Bruch’s membrane from human donors (aged 20–91) was mounted in Ussing chambers to assess hydraulic conductivity. ARPE-19 cells were then cultured on the membrane, and, after forming a confluent monolayer, were removed. Transport measurements were repeated, revealing that RPE proliferation shifted the declining, age-related transport curve upward, thereby improving membrane function (from Ref. [91]).

In vivo, disruption of RPE cells can be accomplished by using sub-micro-second to nano-second pulsed lasers that theoretically restrict heat flow to within 1–3 μm of melanin granules. This thermal confinement around melanosomes leads to microbubble formation with subsequent disintegration of cellular architecture and cell death without any acute effects on Bruch’s membrane or photoreceptors [157,158,159]. In vitro studies have demonstrated that these lasers enhance the release of activated MMPs from human and porcine RPE explants [160,161].

A retinal rejuvenation therapy (2RT) laser system has been developed that utilizes 3 nano-second energy pulses from a Q-switched, frequency-doubled YAG laser at a wavelength of 532 nm (Ellex Pty Ltd., Adelaide, Australia). This system is designed to further reduce pulse energy and selectively target RPE cells without causing damage to the overlying retina or adjacent tissues. In human RPE explant models, laser application led to an approximately sevenfold increase in active-MMP2 levels, which peaked around day 7 and declined by day 18 after treatment. A comparable pattern was observed for active MMP9. Although wound closure was achieved between days 10 and 14, elevated MMP activity persisted for an additional 6 to 8 days after the RPE monolayer had been re-established over the lesion site [160].

Preliminary studies using the 2RT laser system have shown considerable potential as a therapeutic approach for AMD [162,163]. A clinical investigation was conducted as a 36-month, multicentre, randomized, sham-controlled trial. Laser applications (400 μm) were limited to 12 spots per eye, 6 were arranged in an arc positioned below the superior vascular arcade, and the remaining 6 were placed in an arc located above the inferior vascular arcade. Of 222 patients with intermediate AMD who did not exhibit reticular pseudodrusen (RPD), which are deposits in the inter-photoreceptor matrix, progression to advanced AMD was reduced by a factor of four (*p* < 0.002) over the three-year period. These findings are promising and suggest the need for future trials involving a greater number of laser applications, potentially exceeding 100.

Thus, increasing the availability of activated MMP enzymes either by release from trapped sites or by greater activation of pro-MMP2 or from release by the RPE can modulate the ECM of Bruch’s membrane leading to degradation and rejuvenation with subsequent improvement in its functional characteristics.

## 6. Conclusions

The ECM of Bruch’s membrane undergoes marked structural and functional decline with age due to damage initiated in the photoreceptors and RPE and thereafter compounded by extrusion of these toxic metabolites into the ECM of the membrane. Structural changes lead to an increase in membrane thickness due to the accumulation of normal and abnormally damaged ECM components together with a large deposition of lipid-rich proteinaceous debris. This ‘clogging’ of the membrane results in an age-related exponential decline in the transport of nutrients, fluids, and waste products across Bruch’s membrane. In advanced aging of Bruch’s membrane associated with AMD, transport inefficiency delivers a metabolic insult leading to the degeneration of both the RPE and photoreceptors, culminating in blindness.

Bruch’s membrane houses an MMP degradation machinery with the key proteolytic enzyme being active-MMP2. This system operates well in younger subjects, but aging is associated with lowered levels of activated MMP2 that diminishes the degradation potential of Bruch’s membrane. The reduction in degradation is due to both entrapment of active MMP2 within the matrix and the low level of activation of pro-MMP2. The reduced activation of pro-MMP2 results from the limited availability of the free pro-enzyme and restricted mobility of both pro-MMP2 and TIMP2, which impairs their interaction with MMP14 located on the basal surface of the RPE. The reduced pool of free pro-MMP2 arises due to the competing reactions of covalent modification, binding to the matrix, and sequestration into the large LMMC particle.

To date, Bruch’s membrane in AMD has not been extensively examined using zymography, or degradomics to characterize enzymatic activity across disease stages. Such studies are essential to better understand MMP activity and ECM remodeling and should be prioritized in future investigations.

Therapeutic measures that can improve degradation capacity are currently being investigated. Saponins are tri-terpenoid surfactant molecules that have been shown to remove the debris in Bruch’s membrane by releasing trapped active-MMP2 enzymes and improving the diffusional status of the membrane, allowing greater mobility of pro-MMP2 and TIMP2 for activation at the RPE surface. These structural improvements translate into enhanced hydraulic conductivity and diffusion across Bruch’s membrane, facilitating the transport of essential molecules such as retinol. In clinical studies, this has been associated with restoration of dark adaptation kinetics in AMD patients, highlighting the functional impact of saponin treatment. The observation that proliferating RPE cells release substantial amounts of activated-MMP2 and -MMP9 enzymes has resulted in the development of a laser technique to increase the level of activated-MMP enzymes in Bruch’s membrane. Lasers are used to cause small lesions in the RPE layer and as RPE cells surrounding the lesion migrate to close the wound, they release activated MMPs into Bruch’s membrane. Both therapeutic strategies have undergone clinical trials showing great promise as potential treatments for AMD.

## Figures and Tables

**Figure 1 biomolecules-15-01059-f001:**
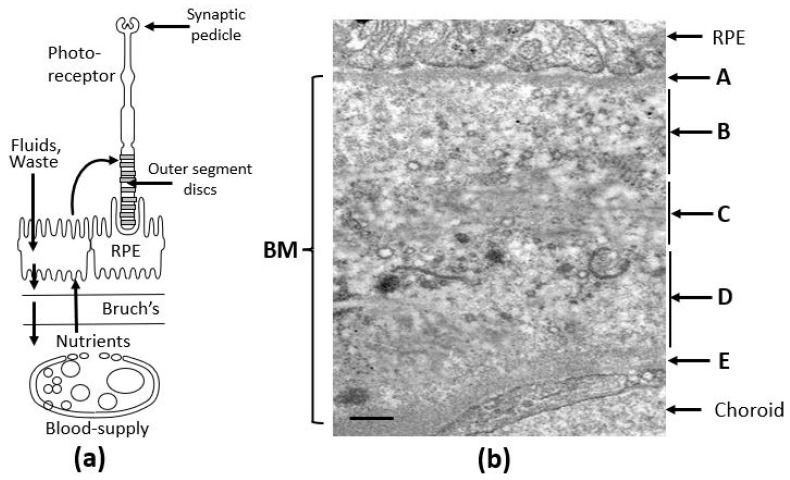
The visual unit of the human eye. (**a**) Situated between the RPE and the choroidal blood supply, Bruch’s membrane serves as a crucial transport interface, mediating the passage of both free and carrier-bound nutrients, antioxidants, and trace metals to the photoreceptor–RPE complex. It also functions as the final barrier for the efflux of fluids, metabolic waste, and membranous debris. The integrity of these transport routes is vital for maintaining photoreceptor and RPE cell survival. (**b**) Electron microscopy image of Bruch’s membrane from human tissue, revealing its distinct penta-laminar organization [17,18]. BM, Bruch’s membrane; RPE, retinal pigment epithelium; A, basement membrane of the RPE; B, inner collagenous layer; C, elastin layer; D, outer collagenous layer; E, basement membrane of the choriocapillaris. Bar marker, 0.5 µm.

**Figure 2 biomolecules-15-01059-f002:**
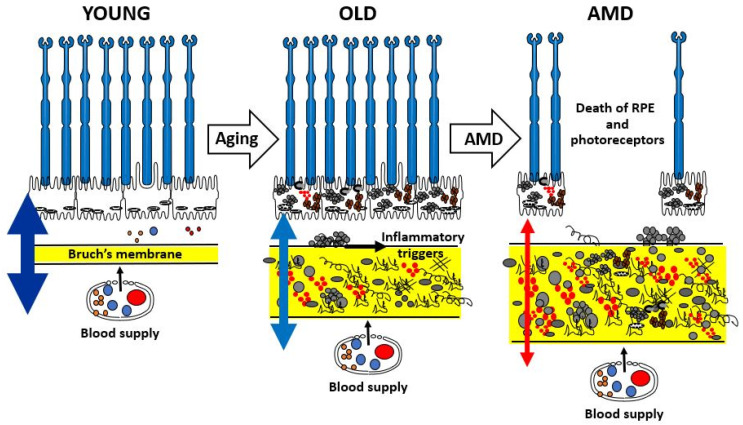
Schematic to show the age-related changes in the structure and function of Bruch’s membrane highlighting the risk of transition to pathology in the elderly. AMD, age-related macular degeneration; RPE, retinal pigment epithelium.

**Figure 3 biomolecules-15-01059-f003:**
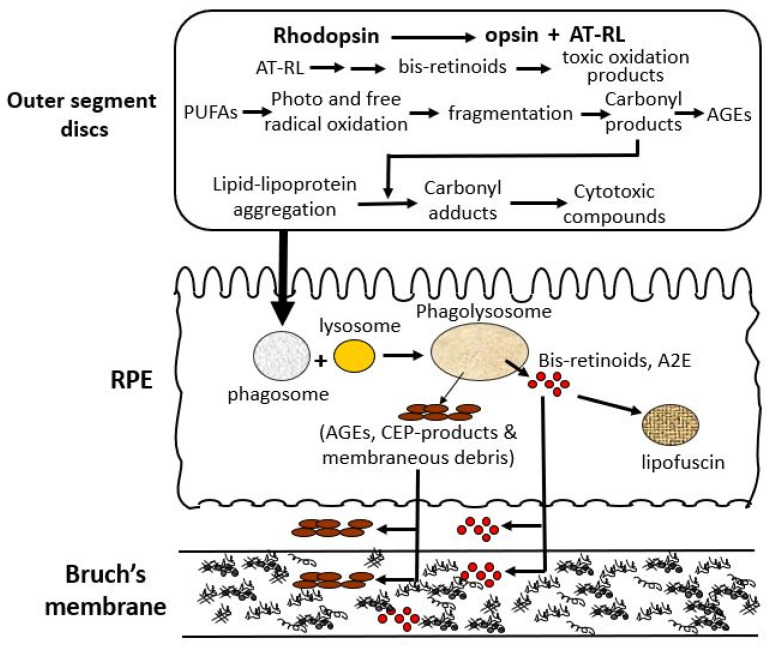
Oxidative stress-induced structural and functional alterations in aging Bruch’s membrane. Photoreceptors generate toxic byproducts including bis-retinoids [33,34,35,36], lipid–protein aggregates [37,38,39,40], and reactive carbonyl compounds [37,38,41,42] through rhodopsin activation, photooxidation of PUFAs [43], and lipid peroxidation [37,38,40,44]. These metabolites undergo limited degradation in the RPE and accumulate as lipofuscin [45,46,47,48] or are deposited onto Bruch’s membrane [11,40,44,49,50,51,52]. Over time, this leads to ECM disruption, impaired transport, and structural degeneration of Bruch’s membrane [10,12,53,54,55,56], contributing to age-related pathology. A2E, N-retinylidene-N-retinylethanolamine; AGEs, advanced glycation end products; AT-RL, all-trans retinal; CEP, carboxyethylpyrrole; ECM, extracellular matrix; PUFAs, polyunsaturated fatty acids; RPE, retinal pigment epithelium.

**Figure 4 biomolecules-15-01059-f004:**
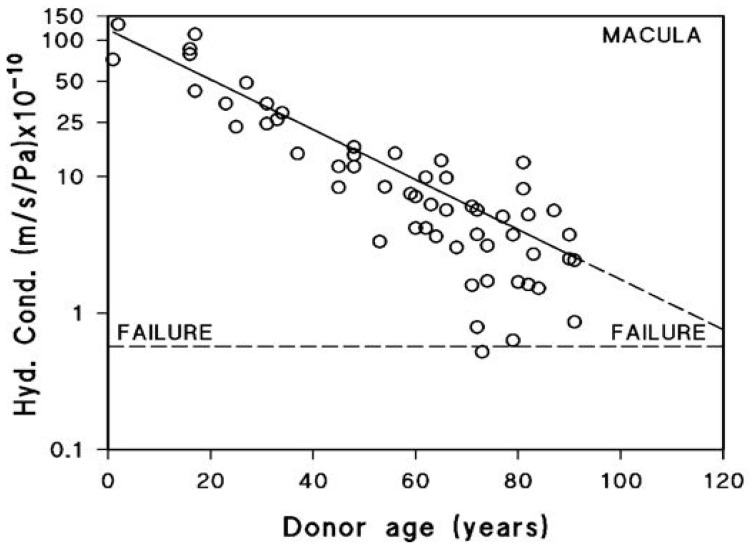
Fluid transport across aging human Bruch’s membrane. Hydraulic conductivity of Bruch’s membrane declines exponentially with age, as shown in 56 donors aged 9 to 91. Older individuals, including some healthy controls, approach the failure threshold, increasing the risk of fluid buildup and RPE detachment. AMD donors show even lower conductivity, suggesting accelerated membrane dysfunction (modified from Refs. [71,73,74]).

**Figure 5 biomolecules-15-01059-f005:**
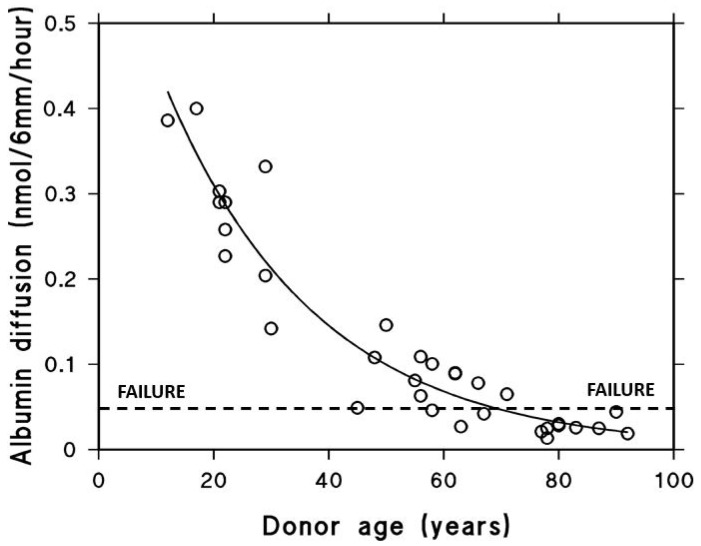
Albumin diffusion across aging human Bruch’s membrane. Diffusion decreases exponentially with age, reaching a functional threshold in the elderly (modified from Ref. [76]).

**Figure 6 biomolecules-15-01059-f006:**
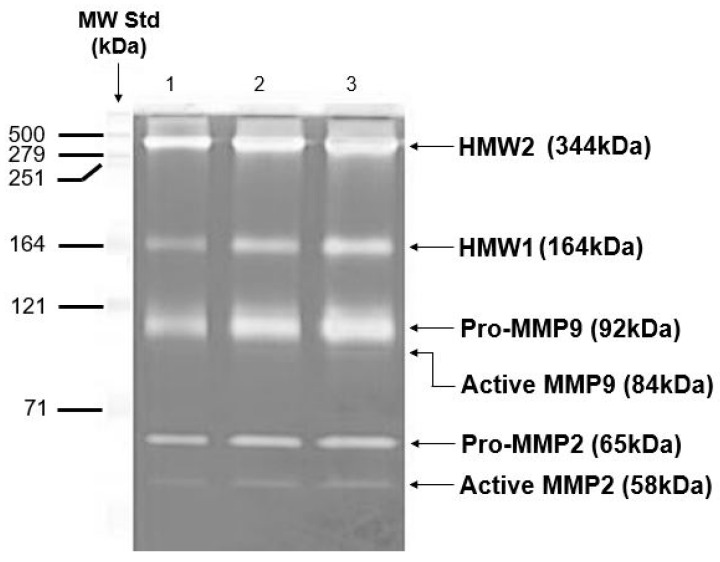
Zymogram to show the various MMP species present in intact human Bruch’s membrane. Zymogram showing pro-MMP2, pro-MMP9, and HMWs in human Bruch’s membrane (adapted from Refs. [97,101,118,119,120,121,122,123]). HMWs, high molecular weight gelatinase species.

**Figure 7 biomolecules-15-01059-f007:**
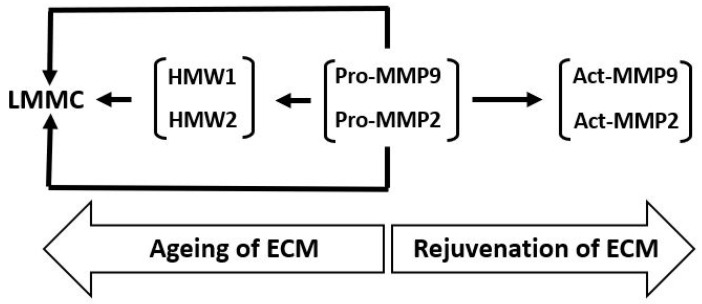
The MMP pathway of aging and rejuvenation of Bruch’s membrane. LMMC, large macromolecular weight MMP complex; HMWs, high molecular weight gelatinase species.

**Figure 8 biomolecules-15-01059-f008:**
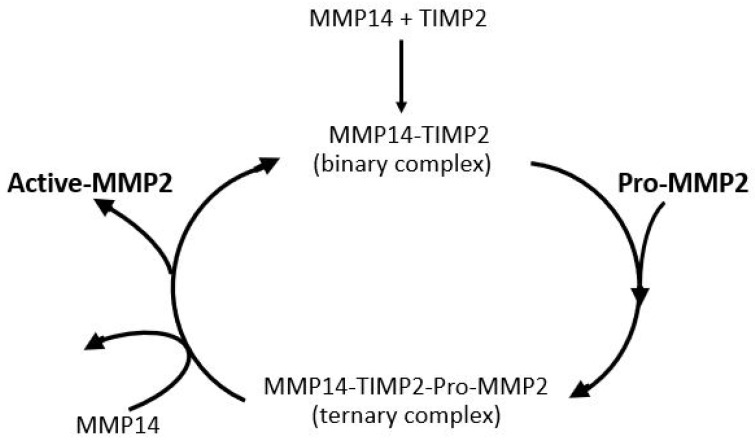
Activation of pro-MMP2. Pro-MMP2 is activated on the RPE basolateral surface through a complex with MMP14 and TIMP2 (adapted from Refs. [138,139,140]).

**Figure 9 biomolecules-15-01059-f009:**
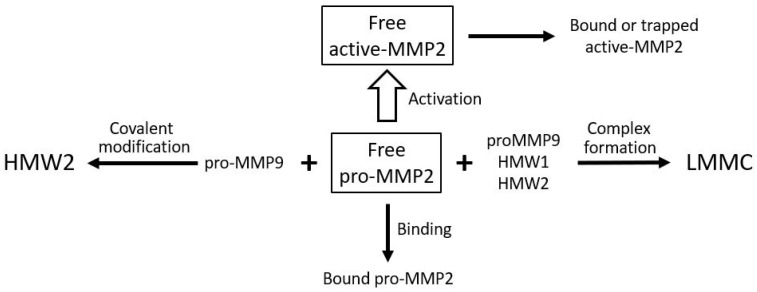
Competing reactions reducing the level of free pro-MMP2 for activation in Bruch’s membrane. Free pro-MMP2 is reduced by matrix binding, covalent complex formation with pro-MMP9 (HMW2), and incorporation into LMMC particles. In AMD, elevated pro-MMP9 accelerates these reactions, limiting pro-MMP2 activation (adapted from Refs. [123,141]).

**Figure 10 biomolecules-15-01059-f010:**
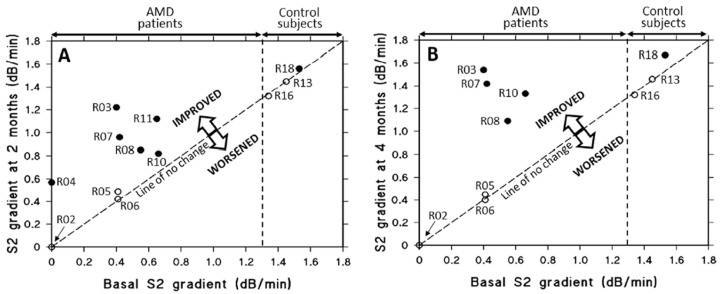
Effect of oral saponin or placebo supplementation on rod S2 gradients over two to four months in AMD patients and controls. (**A**) After two months, placebo-treated subjects (three AMD patients and two controls, open circles) showed little or no change in S2 gradients. In contrast, all AMD patients who received saponins (filled circles) showed improvement (*p* < 0.005). (**B**) At four months, S2 gradients in the placebo group remained unchanged. Four AMD patients who continued saponin treatment showed further improvement compared to their two-month values (*p* < 0.01) (adapted from Ref. [156]). AMD, age-related macular degeneration.

**Figure 11 biomolecules-15-01059-f011:**
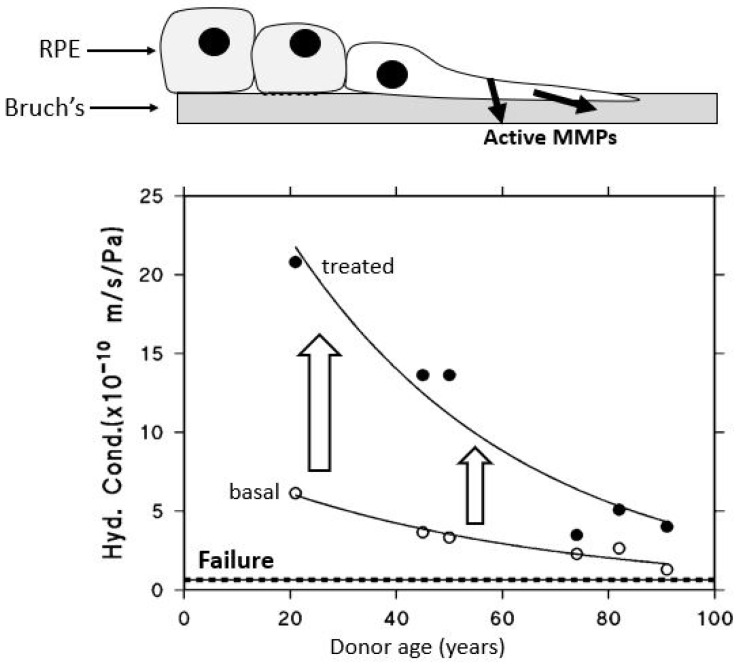
Effect of RPE proliferation on the fluid transporting characteristics of underlying Bruch’s membrane. RPE proliferation enhances the hydraulic conductivity of aged human Bruch’s membrane, likely via MMP activation. In vitro, ARPE-19 cells seeded onto donor membranes improved transport properties after reaching confluence (adapted from Ref. [91]).

## Data Availability

Not applicable.

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
