# Peer review of "Extracellular Matrix (ECM) Aging in the Retina: The Role of Matrix Metalloproteinases (MMPs) in Bruch’s Membrane Pathology and Age-Related Macular Degeneration (AMD)"

_biomolecules, 2025, doi:10.3390/biom15081059_

Round 1
Reviewer 1 Report
Comments and Suggestions for Authors
I appreciate the opportunity to review the manuscript: I appreciate the opportunity to review the manuscript: „ Matrix metalloproteinase (MMP)-mediated remodelling of the extracellular matrix (ECM) in normal aging and age-related diseases.
Recommendation: Minor revision
General comments:
In my opinion, the manuscript is interesting, but requires additions and corrections.
The manuscript is very interesting and presents a high level of substantive preparation. However, the abstract, which does not reflect the content of the work, requires correction. In my opinion, it should be improved.
Strong points of the paper: topicality of the subject, the social utility, especially because the aging of many societies and the pathology associated with age-related macular degeneration (AMD).
Author Response
Comment 1:
The manuscript is very interesting and presents a high level of substantive preparation. However, the abstract, which does not reflect the content of the work, requires correction. In my opinion, it should be improved.
Response1:
We sincerely thank Reviewer 1 for taking the time to evaluate our manuscript titled “Matrix metalloproteinase (MMP)-mediated remodelling of the extracellular matrix (ECM) in normal aging and age-related diseases” and for the positive comments on its topical relevance and scholarly content.
We appreciate your recommendation for minor revision and we have revised the abstract to better align with the overall content, highlighting the key themes and conclusions of the review, particularly the mechanistic role of MMPs in both physiological aging and pathological conditions such as age-related macular degeneration (AMD).
Revisions made:
•
The abstract has been rewritten to more clearly reflect the structure and content of the manuscript.
•
We ensured that the revised abstract emphasizes the societal and clinical importance of ECM remodeling in aging and AMD, as highlighted in your review.
We are grateful for your thoughtful suggestions and believe the changes made have improved the clarity and coherence of the manuscript.
Reviewer 2 Report
Comments and Suggestions for Authors
Please see file.
Congratulations on this fine review!
Regards.

Author Response
Please review the response in the attached file.

Reviewer 3 Report
Comments and Suggestions for Authors
The review titled: “Matrix metalloproteinase (MMP)-mediated remodelling of the extracellular matrix (ECM) in normal aging and age-related diseases ” reviews the alteration in rejuvenation potential of the extracellular matrix due to the decreased degraded capacity of matrix metalloproteinases during the aging process and associated macular degeneration.
Since the review exclusively considers macular degeneration an aging-related disease and cites it as a good model for studying aging processes in the context of deregulation of matrix metalloprotease activity and extracellular matrix renewal, I would advise the authors to emphasize this in the title.
The reference to the article from which the figure is cited must be indicated in the subfigure text.
In many places in the text, changes in MMP and extracellular matrix proteins are discussed in general, rather than the specifics of their interaction and pathology in macular degeneration, and more specifically in Bruch’s membrane.
The text often contains sentences with textbook content, such as: „Electrophoresis separates MMPs according to molecular 376 weight, with the smallest running towards the bottom of the gel. „: Zymographic analysis of the pellet fraction allows quantification of bound MMP species. The supernatant fraction (housing the free MMP pool) 387 can be subjected to gel-filtration column chromatography to separate individual MMP components in their native configuration. „ and others, which is not appropriate for review content.
My opinion is that the review should be rewritten with an attempt to increase the scientific soundness and comprehensiveness of the subject under consideration.
Comments on the Quality of English LanguageThe English should be improved.
Author Response
We thank Reviewer 3 for the constructive comments, which helped us improve the focus and clarity of the manuscript.
In response:
Comments 1:
Since the review exclusively considers macular degeneration an aging-related disease and cites it as a good model for studying aging processes in the context of deregulation of matrix metalloprotease activity and extracellular matrix renewal, I would advise the authors to emphasize this in the title.Response 1:
Title revised: As suggested, we have revised the title to better reflect the central focus on age-related macular degeneration (AMD): “Extracellular Matrix (ECM) Aging in the Retina: The Role of Matrix Metalloproteinases (MMPs) in Bruch’s Membrane Pathology and Age-Related Macular Degeneration (AMD).”
Comments 2:
The reference to the article from which the figure is cited must be indicated in the subfigure text.
Response 2:
Figure reference added: The source of the figure has now been clearly indicated in the figure legend.
Comments 3:
In many places in the text, changes in MMP and extracellular matrix proteins are discussed in general, rather than the specifics of their interaction and pathology in macular degeneration, and more specifically in Bruch’s membrane.
Response 3:
Improved specificity: We have revised sections that previously discussed MMP and ECM changes in general terms, to focus more specifically on their relevance to AMD and Bruch’s membrane pathology.
Comments 4:
The text often contains sentences with textbook content, such as: „Electrophoresis separates MMPs according to molecular 376 weight, with the smallest running towards the bottom of the gel. „: Zymographic analysis of the pellet fraction allows quantification of bound MMP species. The supernatant fraction (housing the free MMP pool) 387 can be subjected to gel-filtration column chromatography to separate individual MMP components in their native configuration. „ and others, which is not appropriate for review content.
My opinion is that the review should be rewritten with an attempt to increase the scientific soundness and comprehensiveness of the subject under consideration.
Response 4:
Textbook-like content revised: Descriptive methodological explanations that were too basic for a review article have been replaced with concise and context-relevant discussion.
We appreciate the reviewer’s input and believe these changes have strengthened the manuscript.
Reviewer 4 Report
Comments and Suggestions for Authors
Dear Authors,
I need to present my comments concerning your paper:
- It seems to be necessary to modify the title of paper by adding the therms showing that the text presents the problem of aging of retina and pathology in Bruchs membrane space. The presented title is not compatible to the content of paper.
- It is necessary to add a small chapter regarding to the role of miRNA in the regulation if axis ECM-MMPs/activity-neovascularisation-AMD.
I have no comments concerning English Language.
Author Response
We sincerely thank Reviewer 4 for the constructive comments on our manuscript.
Comments 1:
It seems to be necessary to modify the title of paper by adding the therms showing that the text presents the problem of aging of the retina and pathology in Bruchs membrane space. The presented title is not compatible to the content of paper.
Response 1:
Title modification: We have revised the title to better reflect the focus on retinal aging and Bruch’s membrane pathology, as suggested: “Extracellular Matrix (ECM) Aging in the Retina: The Role of Matrix Metalloproteinases (MMPs) in Bruch’s Membrane Pathology and Age-Related Macular Degeneration (AMD).”
Comments 2:
It is necessary to add a small chapter regarding to the role of miRNA in the regulation if axis ECM-MMPs/activity-neovascularisation-AMD.
Response 2:
Addition of miRNA section: In line with your recommendation, we have added a paragraph addressing the role of microRNAs (miRNAs) in regulating the ECM–MMPs axis and their relevance to neovascularization and AMD pathogenesis. The following paragraph has been added to the manuscript:
“In addition to enzymatic regulation, microRNAs (miRNAs) act as post-transcriptional modulators of MMPs, influencing ECM remodeling. For example, miR-21 has a protective effect on angiogenesis by reducing cell death and promoting cell survival and migration, in part through targeting TIMP3 and regulating MMP2 and MMP9 (Hu et al., 2016). miR-29b has been identified as a potential regulator of MMP2 (Schellinger et al., 2021). Other miRNAs, such as miR-204, are also involved in RPE homeostasis (Intartaglia et al., 2021). Moreover, modulation of specific miRNAs—such as overexpression of miR-21, miR-31, or miR-150, or silencing of miR-23 and miR-27—has been shown to attenuate Choroidal Neovascularization (CNV) in laser-induced models (Wang et al., 2012). These findings suggest that miRNAs function as endogenous regulators of MMP-dependent ECM remodeling in AMD pathogenesis.”
We are grateful for your insights, which have enhanced the completeness of the manuscript.